# Uncertainty Estimation via Discrete Latent Representation

## Abstract

Many important problems in the real world don't have unique solutions. It is thus important for machine learning models to be capable of proposing different plausible solutions with meaningful probability measures. In this work we propose a novel deep learning based framework, named *modal uncertainty estimation* (MUE), to learn the one-to-many mappings between the inputs and outputs, together with faithful uncertainty estimation. Motivated by the multi-modal posterior collapse problem in current conditional generative models, MUE uses a set of discrete latent variables, each representing a latent mode hypothesis that explains one type of input-output relationship, to generate the one-to-many mappings. Benefit from the discrete nature of the latent representations, MUE can estimate any input the conditional probability distribution of the outputs effectively. Moreover, MUE is efficient during training since the discrete latent space and its uncertainty estimation are jointly learned. We also develop the theoretical background of MUE and extensively validate it on both synthetic and realistic tasks. MUE demonstrates (1) significantly more accurate uncertainty estimation than the current state-of-the-art, and (2) its informativeness for practical use.

## 1 Introduction

Making predictions in the real world has to face with various uncertainties. One of the arguably most common uncertainties is due to partial or corrupted observations, as such it is often insufficient for making a unique and deterministic prediction. For example, when inspecting where a single CT scan of a patient contains lesion, without more information it is possible for radiologists to reach different conclusions, as a result of the different hypotheses they have about the image. In such an ambiguous scenario, the question is thus, given the observable, which one(s) out of the many possibilities would be more reasonable than others? Mathematically, this is a one-to-many mapping problem and can be formulated as follows. Suppose the observed information is $\mathbf{x} \in \mathcal{X}$ in the input space, we are asked to estimate the conditional distribution $p(\mathbf{y}|\mathbf{x})$ for $\mathbf{y} \in \mathcal{Y}$ in the prediction space, based on the training sample pairs $(\mathbf{x}, \mathbf{y})$.

There are immediate challenges that prevent $p(\mathbf{y}|\mathbf{x})$ being estimated directly in practical situations. First of all, both $\mathcal{X}$ and $\mathcal{Y}$, *e.g.*as spaces of images, can be embedded in very high dimensional spaces with very complex structures. Secondly, only the unorganized pairs $(\mathbf{x}, \mathbf{y})$, *not* the one-to-many mappings $\mathbf{x} \mapsto \{\mathbf{y}_i\}_i$, are explicitly available. Fortunately, recent advances in conditional generative models based on Variational Auto-Encoder (VAE) framework from Kingma & Welling (2014) shed light on how to tackle our problem. By modelling through latent variables $\mathbf{c} = \mathbf{c}(\mathbf{x})$, one aims to explain the underlying mechanism of how $\mathbf{y}$ is assigned to $\mathbf{x}$. And hopefully, variation of $\mathbf{c}$ will result in variation in the output $\hat{\mathbf{y}}(\mathbf{x}, \mathbf{c})$, which will approximate the true one-to-many mappings distributionally.

Many current conditional generative models, including cVAE in Sohn et al. (2015), BiCycleGAN in Zhu et al. (2017b), Probabilistic U-Net in Kohl et al. (2018), *etc.*, are developed upon the VAE framework, with Gaussian distribution with diagonal covariance as the *de facto* parametrization of the latent variables. However, in the following we will show that such a parametrization put a dilemma between model training and actual inference, as a form of what is known as the *posterior collapse* problem in the VAE literature Alemi et al. (2018); Razavi et al. (2018). This issue is particularly easy to understand in our setting, where we assume there are multiple $\mathbf{y}$'s for a given $\mathbf{x}$.

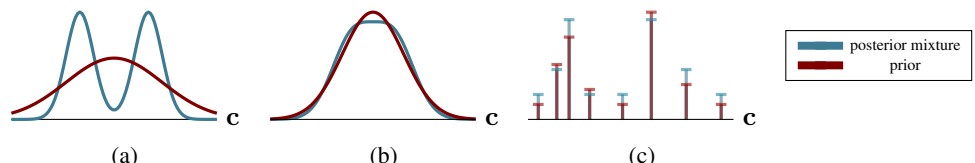

Figure 1: Comparison between Gaussian latent representations and discrete latent representations in a multi-modal situation. Gaussian latents are structurally limited in such a setting. (a) The ideal situation when there is no posterior collapse as multiple modes appear, but the prior distribution is a poor approximation of the posterior. (b) Posterior collapse happens, and no multi-modal information is conveyed from the learned prior. (c) Discrete latent representation can ameliorate the posterior collapse problem while the prior can approximate the posterior more accurately when both are restricted to be discrete.

Let us recall that one key ingredient of the VAE framework is to minimize the KL-divergence between the latent prior distribution $p(\mathbf{c}|\mathbf{x})$ and the latent variational approximation $p_\phi(\mathbf{c}|\mathbf{x}, \mathbf{y})$ of the posterior. Here $\phi$ denotes the model parameters of the "recognition model" in VAE. It does not matter if the prior is fixed $p(\mathbf{c}|\mathbf{x}) = p(\mathbf{c})$ Kingma & Welling (2014) or learned $p(\mathbf{c}|\mathbf{x}) = p_\theta(\mathbf{c}|\mathbf{x})$ Sohn et al. (2015), as long as both prior and variational posterior are parameterized by Gaussians. Now suppose for a particular $\mathbf{x}$, there there are two modes $\mathbf{y}_1, \mathbf{y}_2$ for the corresponding predictions. Since the minimization is performed on the entire training set, $p(\mathbf{c}|\mathbf{x})$ is forced to approximate a *posterior mixture* $p(\mathbf{c}|\mathbf{x}, \mathbf{y}_{(\cdot)})$ of two Gaussians from mode $\mathbf{y}_1$ and $\mathbf{y}_2$. In the situation when the minimization is successful, meaning the KL divergence is small, the mixture of the variational posteriors must be close to a Gaussian, *i.e.*posterior collapsed as in Fig.1(b), and hence the multi-modal information is lost. Putting it in contrapositive, if multi-modal information is to be conveyed by the variational posterior, then the minimization will not be successful, meaning higher KL divergence. This may partly explain why it can be a delicate matter to train a conditional VAE. The situation is schematically illustrated in Figure 1 in one dimension. Note that the case in Figure 1(a) is usually more preferable, however the density values of the prior used during testing *cannot reflect the uncertainty level of the outputs*. We quantitative demonstrate this in Section 4 and Fig.2.

One direction to solve the above problem is to modify the strength of KL-divergence or the variational lower bound, while keeping the Gaussian parametrization, and has been explored in the literature extensively, as in Higgins et al. (2017); Alemi et al. (2018); Rezende & Viola (2018). However, besides the need of extensive parameter tuning for these approaches, they are not tailored for the multi-modal posterior collapse problem we described above, thus do not solve the inaccurate uncertainty estimation problem. Mixture or compositions of Gaussian priors have also been proposed in Nalisnick et al. (2016); Tomczak & Welling (2018), but the number of Gaussians in the mixture is usually fixed apriori. Hence making it a conditional generative model further complicates the matter, since the number in the mixture should depend on the input. We therefore adopt another direction, which is to use a latent distribution parameterization other than Gaussians, and one that can naturally exhibit multiple modes. The simplest choice would be to constrain the latent space to be a finite set, as proposed in van den Oord et al. (2017), so that we can learn the conditional distribution as a categorical distribution.

We argue that the approach of discrete latent space may be beneficial particularly in our setting. First, different from unconditional or weak conditional generative modelling tasks where diversity is the main consideration, making accurate predictions based on partial information often leads to a significantly restricted output space. Second, there is no longer noise injection during training, so that the decoder can utilize the information from the latent variable more effectively. This makes it less prone to ignore the latent variable completely, in contrast to many conditional generation methods using noise inputs. Third, the density value learned on the latent space is more interpretable, since the learned prior can approximate the variational posterior better. In our case, the latent variables can now represent latent mode hypotheses for making the corresponding most likely predictions. We call our approach *modal uncertainty estimation (MUE)*.

The main contributions of this work are: (1) We solve the MUE problem by using c-VAE and justify the use of a discrete latent space from the perspective of multi-modal posterior collapse problem. (2) Our uncertainty estimation improves significantly over the existing state-of-art. (3)

In contrast to models using noise inputs that require sampling at the testing stage, our model can directly produce results ordered by their latent mode hypothesis probabilities, and is thus more informative and convenient for practical use.

The rest of paper is organized as follows. In Section 2 we sample some works that related to ours and stress the key differences between them. In Section 3 we layout our general framework and model details. We conducted a series of experiments on both synthetic and real datasets described in Section 4. The paper is concluded in Section 5.

## 2 RELATED WORK

Conditional generative models aim to capture the conditional distribution of the data and generate them according to some given information. Thanks to the recent advancement of deep learning techniques, especially the methods of generative adversarial networks (GANs) Goodfellow et al. (2014) and variational auto-encoders (VAEs) Kingma & Welling (2014), conditional generative models have been effectively applied to various computer vision and graphics tasks such as image synthesis, style transfer, image in-painting, *etc*. Early works in this direction focused on learning the uni-modal mapping, as in Isola et al. (2017) and Zhu et al. (2017a). They are called uni-modal because the mapping is between fixed categories, namely a one-to-one mapping. There are no latent codes to sample from, thus the generation is deterministic. In these works, images of a specific category are translated to another category, while keeping the desired semantic content. These methods achieved the goal through a *meta supervision* technique known as the adversarial loss as in the GAN framework, where one only needs to supply weak supervision for whether the generated image belongs to a certain category or not. Adversarial loss has been known for producing sharp visual look but it alone cannot guarantee faithful distribution approximation, where issues known as *mode collapse* and *mode dropping* often occur for complicated data distribution Srivastava et al. (2017). In Isola et al. (2017) it is noted that additional noise input in the conditional model in fact fails to increase variability in the output. How to ensure good approximation of output distribution for GANs is still an active area of research. Therefore, the above frameworks might not be suitable for approximating the distribution of one-to-many mappings.

Many works have been proposed to extend to the setting of one-to-many mappings by learning disentangled representations, of *e.g.* "content" and "style", and consequently some form of auto-encoding has to be used. Conditional generation can then be accomplished by corresponding latent code sampling and decoding. This includes the approaches of Zhu et al. (2017b); Huang et al. (2018) for multi-modal image-to-image translation, Zheng et al. (2019) for image in-painting, and many others. Since the main objectives of these works are the visual quality and diversity of the outputs, they are usually not evaluated in terms of the approximation quality of the output distribution. One notable exception is Probabilistic U-Net proposed in Kohl et al. (2018), which is based on the conditional VAE framework Sohn et al. (2015) and is close in spirit to ours. Probabilistic U-Net has shown superior performance over various other methods for calibrated uncertainty estimation, including the ensemble methods of Lakshminarayanan et al. (2017), multi-heads of Rupprecht et al. (2017); Ilg et al. (2018), drop-out of Kendall et al. (2015) and Image2Image VAE of Zhu et al. (2017b). However, as discussed in Section 1, Probabilistic U-Net cannot solve the multi-modal posterior collapse problem since it uses Gaussian latent parameterization. Therefore, in case the conditional distribution is varying for different input data, the performance is expected to degrade. Furthermore, the latent prior density learned has no interpretation, and thus cannot rank its prediction. To perform uncertainty estimation for Probabilistic U-Net, one must perform extensive sampling and clustering.

Our framework improves significantly upon Probabilistic U-Net by introducing discrete latent space. With this latent parameterization we can directly output the uncertainty estimation and we can rank our predictions easily. The discrete latent space has been proposed in the vq-VAE framework of van den Oord et al. (2017). With such a latent space it can get rid of the noise sampling, which enables the latent variable to be more effectively utilized by the decoder and produce outputs with better visual quality. While our use of discrete latent space is motivated by the multi-modal posterior collapse problem. The major technical difference compared to our framework is that the image in vq-VAE framework is encoded by a collection of codes arranged in the spatial order. As such, the joint distribution of the codes cannot be obtained directly, and has to be estimated or sampled using *e.g.* an auto-regressive model in the spatial dimension, such as PixelCNN Van den Oord et al. (2016).

In contrast, we learn disentangled representations and only the necessary information to produce different outputs goes into the discrete latent space. In particular, we model the each mode of $\mathbf{y}$ given $\mathbf{x}$ by a single latent code, thus our model enjoys much simpler sampling.

Besides vq-VAE van den Oord et al. (2017), the use of discrete latent variables in neural network has been explored in various previous works, including the early work of Mnih & Gregor (2014) and Mnih & Rezende (2016) that use single or multiple samples objectives with variance reduction techniques to help training. Others have explored using continuous approximations to the discrete distributions, know as Concrete Maddison et al. (2016) or Gumbel-Softmax Jang et al. (2016) distributions. As is noted in van den Oord et al. (2017), in general the above approaches have fallen short of their continuous counterparts. Worth mentioning is a recently proposed neural dialogue generation method Zhao et al. (2018) that uses Gumbel-Softmax approximation, which treats the dialogue generation as a one-to-many mapping problem. Our method diverge from theirs by the assumption about the model. In Zhao et al. (2018), they designed the learned discrete representation for an utterance to be "context free". This is in contrast to our assumption that the latent hypothesis of an input should depend on the input itself. Taking the task of medical image segmentation for an example, if we encode the hypotheses from the segmentation alone as in Zhao et al. (2018), likely there will either be two modes (benign vs malignant) or a huge number of modes if the shape of the segmentation is taken into account. Moreover, it will not contain any information about what kinds of actual biological tissue they might be, which on the other hand can be judged from the actual scan image. In our case, we have deliberately separated the recognition task learning, e.g. segmenting the image, and the hypothesis learning, so that together they can approximate the variation of the outputs given the input.

Finally, we briefly summarize the differences between MUE and existing uncertainty estimation methodologies in deep learning. Many existing works Gal & Ghahramani (2016); Gal (2016); Kendall et al. (2015); Kendall & Gal (2017) focus on *model uncertainty*, which try to capture the calibrated level of confidence of the model prediction by using stochastic regularization techniques. Such uncertainty will be of major interest for model predictions on unseen data and long-tail rare cases, or when model is trained on limited data. While ours is more about learning from conflicting or ambiguous training data, and estimating the calibrated uncertainty of the input-output relationship in the dataset. Interestingly, Kohl et al. (2018) has experimented using Dropout as comparison to the c-VAE framework in the MUE setting, but found it only achieved inferior performance. In general, since MUE is independent from the model uncertainty, our framework can be used jointly with existing techniques for prediction confidence estimation.

## 3 METHOD

### 3.1 GENERAL FRAMEWORK

Let $(\mathbf{x}, \mathbf{y})$ denote the data-label pair. We model the generation of $\mathbf{y}$ conditioned on $\mathbf{x}$ using the conditional VAE framework as in Sohn et al. (2015). First, a latent variable $\mathbf{c}$ is generated from some prior distribution $p_\theta(\mathbf{c}|\mathbf{x})$ parametrized by a neural network. Then the label $\mathbf{y}$ is generated from some conditional distribution $p_\theta(\mathbf{y}|\mathbf{c}, \mathbf{x})$. We use $\theta$ to represent the collection of model parameters at testing time. The major distinction of our approach is that we assume $\mathbf{c}$ takes value in a finite set $\mathcal{C}$, thought of as a code book for the *latent mode hypotheses* of our multi-modal data distribution. Our goal is to learn the optimal parameters $\theta^*$ and the code book $\mathcal{C}$, so that possibly multiple of the latent modes corresponding to $\mathbf{x}$ can be identified, and label predictions $\hat{\mathbf{y}}$ can be faithfully generated from $\mathbf{x}$. The latter means the marginal likelihood $p_\theta(\mathbf{y}|\mathbf{x})$ should be maximized.

The variational inference approach as in Kingma & Welling (2014) starts by introducing a posterior encoding model $q_\phi(\mathbf{c}|\mathbf{x}, \mathbf{y})$ with parameters $\phi$, which is used only during training. Since the label information is given, we will assume the posterior encoding model is *deterministic*, meaning there is no "modal uncertainty" for the posterior encoding model. So the posterior distribution will be a delta distribution for each data-label pair $(\mathbf{x}, \mathbf{y})$. In any case, the marginal log-likelihood of $\mathbf{y}$ can now be written as

$$\log p_\theta(\mathbf{y}|\mathbf{x}) = \mathbb{E}_{q_\phi(\mathbf{c}|\mathbf{x},\mathbf{y})}\left[\log\frac{q_\phi(\mathbf{c}|\mathbf{x},\mathbf{y})}{p_\theta(\mathbf{c}|\mathbf{x},\mathbf{y})}\right] + \mathbb{E}_{q_\phi(\mathbf{c}|\mathbf{x},\mathbf{y})}\left[\log\frac{p_\theta(\mathbf{c},\mathbf{y}|\mathbf{x})}{q_\phi(\mathbf{c}|\mathbf{x},\mathbf{y})}\right] \tag{1}$$

Since the first term in the RHS of (1) is non-negative, we have the variational lower bound

$$\log p_\theta(\mathbf{y}|\mathbf{x}) \geq \mathbb{E}_{q_\phi(\mathbf{c}|\mathbf{x},\mathbf{y})} \left[ \log \frac{p_\theta(\mathbf{c}, \mathbf{y}|\mathbf{x})}{q_\phi(\mathbf{c}|\mathbf{x},\mathbf{y})} \right]$$
$$= -\mathbb{E}_{q_\phi(\mathbf{c}|\mathbf{x},\mathbf{y})} \left[ \log \frac{q_\phi(\mathbf{c}|\mathbf{y},\mathbf{x})}{p_\theta(\mathbf{c}|\mathbf{x})} \right] + \mathbb{E}_{q_\phi(\mathbf{c}|\mathbf{x},\mathbf{y})} \left[ \log p_\theta(\mathbf{y}|\mathbf{c},\mathbf{x}) \right] \quad (2)$$

We further lower bound Equation (2) by observing that the entropy term $-\mathbb{E}_{q(\cdot)}(\log q(\cdot))$ is positive and is constant if $q_\phi(\mathbf{c}|\mathbf{x}, \mathbf{y})$ is deterministic. This yields a sum of a negative cross entropy and a conditional likelihood

$$\log p_\theta(\mathbf{y}|\mathbf{x}) \geq \mathbb{E}_{q_\phi(\mathbf{c}|\mathbf{x},\mathbf{y})} \left[ \log p_\theta(\mathbf{c}|\mathbf{x}) \right] + \mathbb{E}_{q_\phi(\mathbf{c}|\mathbf{x},\mathbf{y})} \left[ \log p_\theta(\mathbf{y}|\mathbf{c},\mathbf{x}) \right] \quad (3)$$

For our optimization problem, we will maximize the lower bound (3). Since $\mathbf{c}$ takes value in the finite code book $\mathcal{C}$, the probability distribution $p_\theta(\mathbf{c}|\mathbf{x})$ can be estimated using multi-class classification, and the cross entropy term can be estimated efficiently using stochastic approximation.

It is important to note that since we assume $q_\phi(\mathbf{c}|\mathbf{x}, \mathbf{y})$ is deterministic, we will not regularize it by pulling it to the prior distribution, in contrast to the previous conditional VAE frameworks. This means that the *probability values* of the posterior is not influenced by the *probability values* of the prior distribution. Instead, we will let the prior encoding model to be *trained by the posterior encoding model*, as a classification task with ground-truth being the class index obtained from the posterior encoder and the code book $\mathcal{C}$. The lacking of *prior regularization* is also featured in the vq-VAE approach in van den Oord et al. (2017) for unconditional generation, and in Razavi et al. (2018) it is argued that restricting the latent space $\mathcal{C}$ to be a finite set is itself a structural prior constraint for the VAE framework. Note that here the discrete latent space $\mathcal{C}$ should be considered just as a finite set of indices, without any other structure between these indices. Below we will discuss how to realize it in $\mathbb{R}^n$ so that the actual representation can be useful to the decoder.

First of all, because $\mathcal{C}$ is a finite set, the objective (3) is not fully differentiable. We tackle this problem using a simple gradient approximation method and an extra regularization loss, following the approach of van den Oord et al. (2017); Razavi et al. (2019). In details, denote the prior encoding network by $E_\theta$, the posterior encoding network by $E_\phi$, and the decoder as $D_\theta$. Since we assume a delta distribution for the posterior encoding model $q_\phi(\mathbf{c}|\mathbf{x}, \mathbf{y})$, we can let the posterior encoder produce a deterministic output for the given input-output pair $(x, y)$. In other words, no sampling is performed by the posterior encoder. Suppose the output of the posterior encoding network is $\mathbf{e} = E_\phi(\mathbf{x}, \mathbf{y})$. Its nearest neighbor $\mathbf{c}$ in $\mathcal{C}$ in $\ell^2$ distance

$$\mathbf{c} = \arg\min_{\mathbf{c}' \in C} \|\mathbf{c}' - \mathbf{e}\|^2$$

will become the input to the decoder network. And we simply copy the gradient of $\mathbf{c}$ to that of $\mathbf{e}$ so that the posterior encoder can obtain gradient information from the label prediction error. To make sure the gradient approximation is accurate, we need to encourage the posterior encoder's outputs to approximate values in $\mathcal{C}$ as close as possible. To achieve this we use an $\ell^2$-penalization of the form $\beta\|\mathbf{e} - sg[\mathbf{c}]\|^2$ with parameter $\beta > 0$, and $sg$ is the stop-gradient operation. The code $\mathbf{c}$ is updated using exponential moving average of the corresponding posterior encoder's outputs. In the above notation, our loss function to be minimized for a single input pair $(\mathbf{x}, \mathbf{y})$ is

$$\mathcal{L}(\theta, \phi) = CE(E_\theta(\mathbf{x}), id_\mathbf{c}) + Recon(D_\theta(\mathbf{c}, \mathbf{x}), \mathbf{y}) + \beta\|E_\phi(\mathbf{x}, \mathbf{y}) - sg[\mathbf{c}]\|^2 \quad (4)$$

where $CE$ denotes the cross entropy loss, $E_\theta(\mathbf{x})$ is the probability vector of length $|C|$ and $id_\mathbf{c}$ is corresponding code index for the input pair $(\mathbf{x}, \mathbf{y})$. $Recon$ denotes the label reconstruction loss in lieu of the negative log-likelihood.

During training, we learn the prior encoding model $p_\theta(\mathbf{c}|\mathbf{x})$, the posterior encoding model $q_\phi(\mathbf{c}|\mathbf{x}, \mathbf{y})$, the decoding model $p_\theta(\mathbf{y}|\mathbf{c}, \mathbf{x})$, together with the code book $\mathcal{C}$ in an end-to-end fashion. The posterior encoder plus decoder will learn good representation of the latent code, and the prior encoder will learn faithful uncertainty estimation from the stochastic training. At inference time, we use the learned prior encoding model to output a conditional distribution given $\mathbf{x}$ on $\mathcal{C}$, where each of the code will correspond to a decoded label prediction with the associated probability.

## 3.2 MODEL DESIGN AND TRAINING

Our proposed framework in Section 3.1 is general and can be applied to a wide range of network architecture designs. Nonetheless it is worth discussing how the latent code **c** should be learned and utilized, so that the posterior encoding model can really learn the mode hypothesis of the given data-label pair, not simply memorize the data which causes an overwhelming number of codes and over-fitting. One important principle is that the latent code should not be learned or used in a spatially dependent manner, especially in pixel-level recognition tasks such as segmentation. This can also ensure that the prior encoding network is learning to solve the majority of the recognition problem while the latent code supplies only the additional but necessary information to reconstruct distinct outputs from the same (or similar) input. For this purpose we have adopted a simple approach: the code output by posterior encoding network is obtained by global average pooling of its last layer; for the incorporation of the code into the intermediate feature of the decoding network that has spatial dimension $(h, w)$, we will simply make $h \times w$ copies of the code, reshape it to have spatial size $(h, w)$ and concatenate it to the corresponding feature.

In the experiments in Section 4 we will consider applications in computer vision and thus will use convolutional neural networks (CNNs) for the prior and posterior encoder, as well for the decoder, which together is similar to the U-Net architecture. Specifically, each encoding network consists of a sequence of downsampling residual blocks, and the decoder a sequence of upsampling residual blocks, where the decoder also has skip connections to the prior encoder by receiving its feature at each resolution level. The latent code is incorporated at a single level $L$ into the decoder, which depends on the task.

Suppose the latent code is $c$-dimensional. We initialize the code book $\mathcal{C}$ as a size $(n_{\mathcal{C}}, c)$ *i.i.d* random normal matrix, where each column represent an individual code. The statistics of the normal distribution is computed from the output of the posterior encoding network at initialization on the training data. We have found it to be beneficial since it allows the entire model to be initialized at a lower loss position on the optimization landscape. We have also found that the number of code utilized during training follows an interesting pattern: at the very first stage only very few codes will be used, then the number gradually grows to maximum before it slowly declines and becomes stable when the reconstruction loss plateaus. We therefore allow the network to warm up in the first $\eta$ epochs by training without the cross-entropy loss, since during this stage the number of utilized codes is unstable. This will not impair the learning of the posterior encoder, since it receive no gradient information from the prior. We have found $n_{\mathcal{C}} = 512$ to be well sufficient for all of our tasks, and the actual number of codes utilized after training is usually a fraction of it. Because of these observations, we did not try to explicitly enforce different codes to have different outputs, since for one reason the final number of codes are usually compact, and for the other we would like to allow different codes to have similar outputs, which means the possible situation where different hypotheses lead to similar predictions. We expect there will be some connection with the information bottleneck theory Tishby & Zaslavsky (2015) and leave this direction for future work. We will release our open source implementation to promote future research.

## 4 EXPERIMENTS

To rigorously access our method's ability to approximate the distribution of one-to-many mappings, in Section 4.1 we first conduct a synthetic experiment with known ground truth conditional distributions. In Section 4.2 we then demonstrate our method's performance on the realistic and challenging task of lesion segmentation on possibly ambiguous lesion scan on the LIDC-IDRI benchmark. In both experiments we compare with the state-of-the-art method Probabilistic U-Net Kohl et al. (2018) of the same model complexity as ours. More details on the experimental setting and additional results can be found in Appendix A.

### 4.1 QUANTITATIVE ANALYSIS ON SYNTHETIC TASKS

**MNIST guess game**   To test the ability for multi-modal prediction quantitatively, we design a simple guessing game using the MNIST dataset LeCun & Cortes (2010) as follows. We are shown a collection of images and only one of them is held by the opponent. The image being held is not fixed and follows certain probability distribution. We need to develop a generative model to understand the mechanism of which image is being held based on the previously

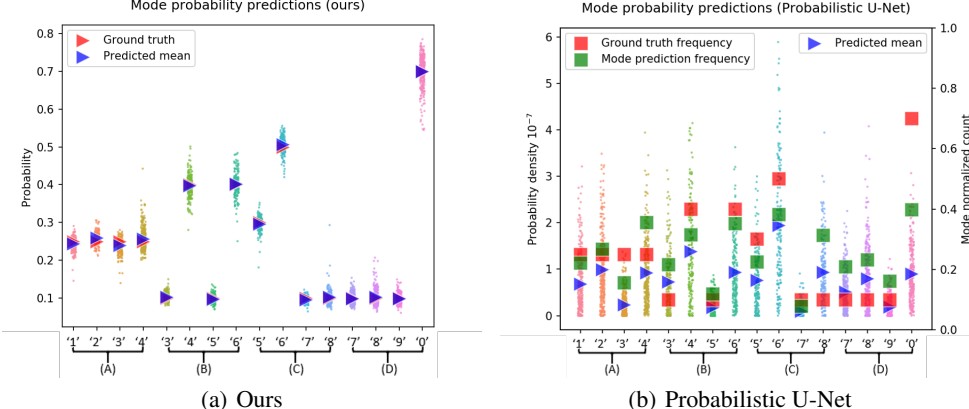

(a) Ours
(b) Probabilistic U-Net

Figure 2: Quantitative comparison on the MNIST guessing task. The small dots represent the predictions for 1000 testing samples. Our method in (a) successfully produces accurate uncertainty estimate for each mode. Probabilistic U-Net uses conventional Gaussian latent parametrization, thus the sample's density provides no useful information about the uncertainty level, as shown in the left axis. We also count the frequencies for each category and plot it on the right axis. However, the approximation is far less accurate than ours even calculated on the entire testing dataset.

seen examples. In details, the input $\mathbf{x}$ will be an image that consists of four random digits, and belongs to one of the four categories: (A) $(1, 2, 3, 4)$; (B) $(3, 4, 5, 6)$; (C) $(5, 6, 7, 8)$; (D) $(7, 8, 9, 0)$. The number represents the label of the image sampled. The output $\mathbf{y}$ will be an image of the same size but only one of the input digit is present. Specifically, for (A) the probability distribution is $(0.25, 0.25, 0.25, 0.25)$; for (B) $(0.1, 0.4, 0.1, 0.4)$; for (C) $(0.3, 0.5, 0.1, 0.1)$; for (D) $(0.1, 0.1, 0.1, 0.7)$. Note that the distribution of the output, conditioned on each category's input, consists of four modes, and is designed to be different for each category. We require the model to be trained *solely* based on the observed random training pairs $(\mathbf{x}, \mathbf{y})$, and thus no other information like digit categories should be used. The model would therefore need to learn to discriminate each category and assign the correct outputs with corresponding probabilities.

Thus for instance, an input image of Category (A) will be the combination of four random samples from Digit $1$ to $4$ in that order, and the output can be the same digit $1$ in the input with probability $0.25$, or it can be the same digit $2$ with probability $0.25$, and so forth. The images in the first row in Fig.3 illustrate an input image, where we also annotate the ground truth probability on the upper-left corner.

We trained our model on samples from the training dataset of MNIST and tested it on samples from the testing dataset, during both stages the random combination is conducted on the fly. The model here for demonstrating the results used a total of 11 codes after training. Please refer to Appendix A.1 for training specifics and more results.

From the second to fifth row in Fig.3 we show the results of different models. Ours in Fig.3(a) are the top-4 predictions with explicit probability estimates annotated on the upper-left in each row. For example, the second row has probability $0.2634$, which is

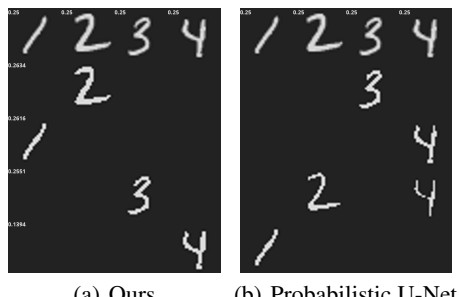

(a) Ours           (b) Probabilistic U-Net

Figure 3: Results visualization on the MNIST guess game. Note that our model can predict calibrated uncertainty estimation (*e.g.* "2" appears with probability $0.25$ versus our prediction $0.2634$, best to zoom on a screen to see the probability annotation in (a)). While Probabilistic U-Net cannot predict such estimates and can produce non-sensible output from random sampling.

very close to the ground truth $0.25$. In contrast, Probabilistic U-Net cannot rank its outputs and hence four random samples are drawn. Consequently one has little control when generating the samples and it's likely to obtain non-sensible outputs as in the fourth row of Fig.3(b).

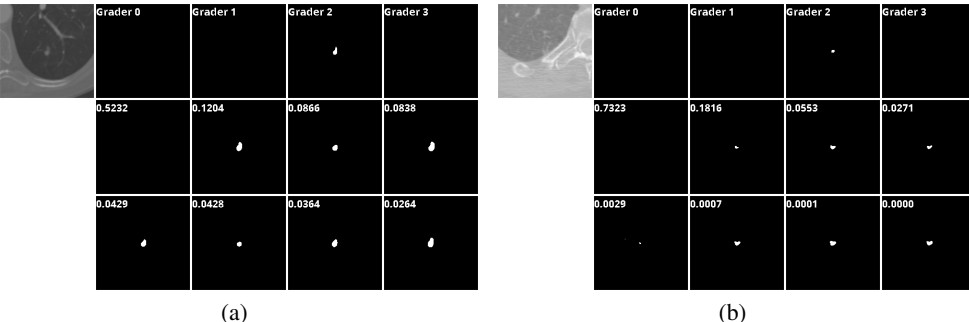

Figure 4: Visualization of our results on the highly ambiguous samples from LIDC-IDRI dataset. The first row shows the input samples and their segmentations, and the next two rows show the top-8 predictions from our method. The uncertainty estimation for each segmentation proposal is annotated on the upper-left corner.

Our method also performs much better quantitatively, as shown in Fig.2 with the results on 1000 random testing samples. We classify both models' outputs into the ground truth modes and aggregate the corresponding probabilities. We can see in Fig.2(a) that our method successfully discovered the distributional properties of the one-to-many mappings, and provides accurate uncertainty estimate. In contrast, due to the Gaussian latent parametrization, neither the individual density of each input nor their averages can provide useful information, as shown by the left axis of Fig.2(b). By the right axis of Fig.2(b) we also count the mode frequencies for each category for Probabilistic U-Net. However, even calculated on the entire testing dataset, the distribution approximation is still far from accurate compared to ours. Note that our method can directly output the uncertainty estimate for *each input* accurately. This clearly demonstrates our method's superiority and practical value.

## 4.2 REAL APPLICATIONS

**Lesion segmentation of possibly ambiguous lung CT scans**   We use the LIDC-IDRI dataset provided by Armato III et al. (2015; 2011); Clark et al. (2013), which contains 1018 lung CT scans from 1010 patients. Each scan has lesion segmentations by four (out of totally twelve) expert graders. The identities of the graders for each scan are unknown from the dataset. Samples from the testing set can be found in the first row of Fig.4. As can be seen, the graders are often in disagreement about whether the scan contains lesion tissue. We hypothesize that the disagreement is due to the different assumptions the experts have about the scan. For example, judging from the scan's appearance, one of the graders might have believed that the suspicious tissue is in fact a normal tissue based on his/her experience, and thus gave null segmentation. There are also other possible underlying assumptions for the graders to come up with different segmentation shapes.

Our task is to identify such ambiguous scenarios by proposing distinct segmentation results from the corresponding latent hypotheses with their associated probabilities, which will be helpful for clinicians to easily identify possible mis-identifications and ask for further examinations of the patients.

Our network architecture for this task is a scaled-up version of the same model used in the MNIST guessing task. At training time, we randomly sample an CT scan $\mathbf{x}$ from the training set, and we randomly sample one of its four segmentations as $\mathbf{y}$. The model we used to report the results has a total of 31 codes. The specifics of the training and more results can be found in Appendix A.3.

Some sample testing results predicted by our model to have *high uncertainty* are illustrated in Fig.4. The first row is the input and its four segmentations, and the last two rows are our top-8 predictions, where the probability associated to each latent code is annotated on the upper-left corner. We can see that our method can capture the uncertainty that is contained in the segmentation labels with notable probability scores, as well as other type of segmentations that seem plausible without further information.

Since no ground truth distribution for LIDC-IDRI dataset is available, quantitative evaluation has to be conducted differently from the MNIST guessing task. We follow the practice of Kohl et al. (2018) to adopt the *generalized energy distance* metric $D^2_{\mathrm{GED}}$ found in Bellemare et al. (2017); Salimans et al. (2018); Székely & Rizzo (2013), which is a statistical quantity that measures the

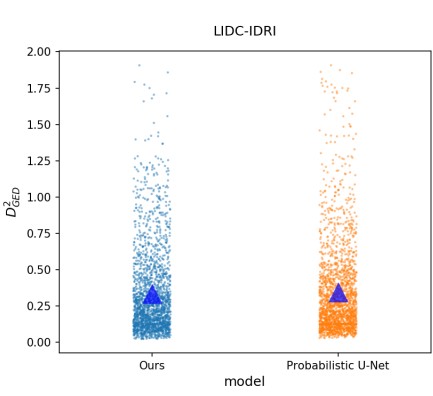

(a) Quantitative comparison on LIDC

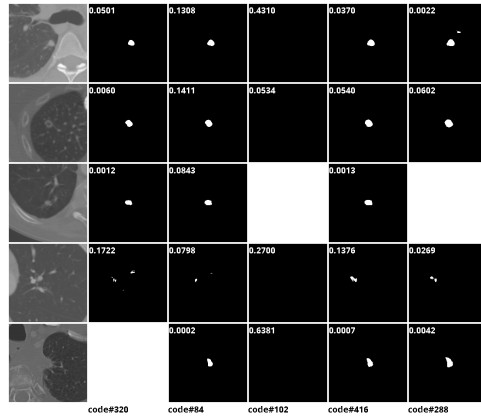

(b) Segmentation-code visualization

Figure 5: Results visualization on LIDC segmentation task. (a) The small dots represent test data instances' $D^2_{\mathrm{GED}}$ values and the triangles mark the mean values. Our performance is competitive with the state-of-the-art. (b) We show the segmentation results for some frequently used code, annotated at the bottom. The segmentation from the codes of probability $< 10^{-4}$ are left blank.

incoherence between two subsets of a metric space. Please refer to Appendix A.2 for calculation details of this metric for segmentations based on Intersection over Union (IoU). The lower the value of $D^2_{\mathrm{GED}}$, the closer the two subsets are. We report the results on the entire testing dataset in Fig.5(a). For our model, the mean $D^2_{\mathrm{GED}}$ of all testing data is $0.3354$, the standard deviation is $0.2947$. Our performance is thus competitive with that of Probabilistic U-Net, whose mean is $0.3470$ and the standard deviation is $0.3139$. Moreover, our model can give quantitative uncertainty estimate directly for each input scan, unlike Probabilistic U-Net that needs to perform sampling and clustering using a metric such as IoU to obtain uncertainty estimate.

Finally, we visualize some segmentation results for some frequently used codes in Fig.5(b). The code used is annotated at the bottom. The segmentations from the codes of negligible probability (*e.g.*less than $10^{-4}$) are left blank. For example, the fourth column for code#102 may correspond to a latent hypothesis that leads to the conclusion of no lesion, and the scan in the third row is not compatible with that particular latent hypothesis. It would be an interesting future work to explore the semantics of the latent codes if more information about the patient and the scan is given.

## 5 DISCUSSION AND CONCLUSION

We have proposed MUE, a novel framework for learning one-to-many mapping with calibrated uncertainty estimation. As an effective solution of the multi-modal posterior collapse problem, the discrete latent representations are learned to explain the corresponding types of input-output relationship in one-to-many mappings. It also allows us effectively and efficiently perform uncertainty estimation of the model prediction. We have extensively validated our method's performance and usefulness on both synthetic and realistic tasks, and demonstrate superior performance over the state-of-the-art methods.

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

APPENDIX

## A  EXPERIMENTAL DETAILS

As described in the main text, each of our encoding network consists of a sequence of downsampling residual blocks, and the decoder a sequence of upsampling residual blocks. The decoder also has skip connections to the prior encoder by receiving its feature at each resolution level. A residual block consists of three convolution layers. As the shortcut connection, the input is added to the output if the input and the output have the same channel size, or otherwise a $1 \times 1$ convolution is applied before the addition. Bi-linear down sampling and up sampling is applied before the residual blocks if the spatial size is changed. We fix the $\ell^2$ penalization weight $\beta = 0.25$, number of initial candidate codes $n_{\mathcal{C}} = 512$, and use the Adam optimizer Kingma & Ba (2015) with its default setting for all of our experiments. Hyper-parameters specific to each experiment are detailed in the following subsections.

### A.1  MNIST GUESSING GAME

We use 6 layers in the prior and the posterior encoding networks, with output channel dimension $[16, 32, 64, 128] + [128, 128]$. This notation means that the first 4 levels are used as in the U-Net which feed the decoder, and the last 2 levels are used for the latent code learning. The posterior encoder further uses a $1 \times 1$ and global average pooling to obtain the code. The code is of dimension 128. The prior encoder uses a linear layer to learn the distribution on $\mathcal{C}$. Our decoder has output channel dimension $[64, 32, 16, 1]$. We incorporate the code at the bottom level, namely the 1-st layer of the decoder. For the Probabilistic U-Net, since the architecture is different, we used a structure of similar capacity, with the parameter *num_filter*= $[16, 32, 64, 128]$ in its released version, and we find the suggested hyperparameters in Kohl et al. (2018) for LIDC task works well in this case. For both networks, we use the binary cross entropy loss, a batch size of 256, and use the learning rate schedule $[1e^{-4}, 5e^{-5}, 1e^{-5}, 5e^{-6}]$ at $[0, 30k, 90k, 120k]$ iterations.

Some additional results from our model and Probabilistic U-Net are shown in Fig.6 and Fig.7.

### A.2  GENERALIZED ENERGY DISTANCE METRIC FOR SEGMENTATIONS

Denote $\mathcal{Y}_\mathbf{x}, \mathcal{S}_\mathbf{x} \subset \mathcal{Y}$ to be the set of segmentation labels and the set of model predictions corresponding to the scan $\mathbf{x}$, respectively. $\mathcal{Y}$ is equipped with the metric

$$d(\mathbf{y}, \mathbf{s}) = 1 - \text{IoU}(\mathbf{y}, \mathbf{s}),$$

where $\text{IoU}(\cdot, \cdot)$ is the intersection-over-union operator that is suitable for evaluating the similarity between segmentations. The $D^2_{\text{GED}}$ statistic in our case is defined to be

$$D^2_{\text{GED}}(\mathcal{Y}_\mathbf{x}, \mathcal{S}_\mathbf{x}) = 2 \sum_{\mathbf{y} \in \mathcal{Y}_\mathbf{x}} \sum_{\mathbf{s} \in \mathcal{S}_\mathbf{x}} p_\mathbf{s} p_\mathbf{y} d(\mathbf{y}, \mathbf{s}) - \sum_{\mathbf{y} \in \mathcal{Y}_\mathbf{x}} \sum_{\mathbf{y}' \in \mathcal{Y}_\mathbf{x}} p_\mathbf{y} p_{\mathbf{y}'} d(\mathbf{y}, \mathbf{y}') - \sum_{\mathbf{s} \in \mathcal{S}_\mathbf{x}} \sum_{\mathbf{s}' \in \mathcal{S}_\mathbf{x}} p_\mathbf{s} p_{\mathbf{s}'} d(\mathbf{s}, \mathbf{s}'),$$

where $p_\mathbf{s}$ is our model's probability prediction for the output $\mathbf{x}$ and $p_\mathbf{y}$ is the ground truth probability. In case the ground truth is not available like LIDC-IDRI, we use $p_\mathbf{y} = \frac{1}{|\mathcal{Y}_\mathbf{x}|}$, where $|\mathcal{Y}_\mathbf{x}|$ denotes the cardinality of $\mathcal{Y}_\mathbf{x}$. For our model, we choose $\mathcal{S}_\mathbf{x}$ to be the top-$N$ predictions. To be rigorous we normalize the sum of their probabilities to be 1 (though which in fact has negligible effect since $N$ are usually chosen so that probability always almost sum up to 1). In the case of Probabilistic U-Net, we use $N$ random output samples and $p_\mathbf{s}$ is replaced by $\frac{1}{|\mathcal{S}_\mathbf{x}|} = \frac{1}{N}$.

### A.3  LIDC-IDRI SEGMENTATION

We use 6 layers in the prior and the posterior encoding networks, with output channel dimension $[32, 64, 128, 192] + [256, 512]$. The code is of dimension 128. Our decoder has output channel dimension $[128, 64, 32, 1]$. We incorporate the code at the bottom level, namely the 1-st layer of the decoder. For the Probabilistic U-Net, since the architecture is different, we used a structure of similar capacity, with the parameter *num_filter*= $[32, 64, 128, 192]$ in its released version, and follows the suggested hyperparameters in Kohl et al. (2018). For both networks, we use the binary

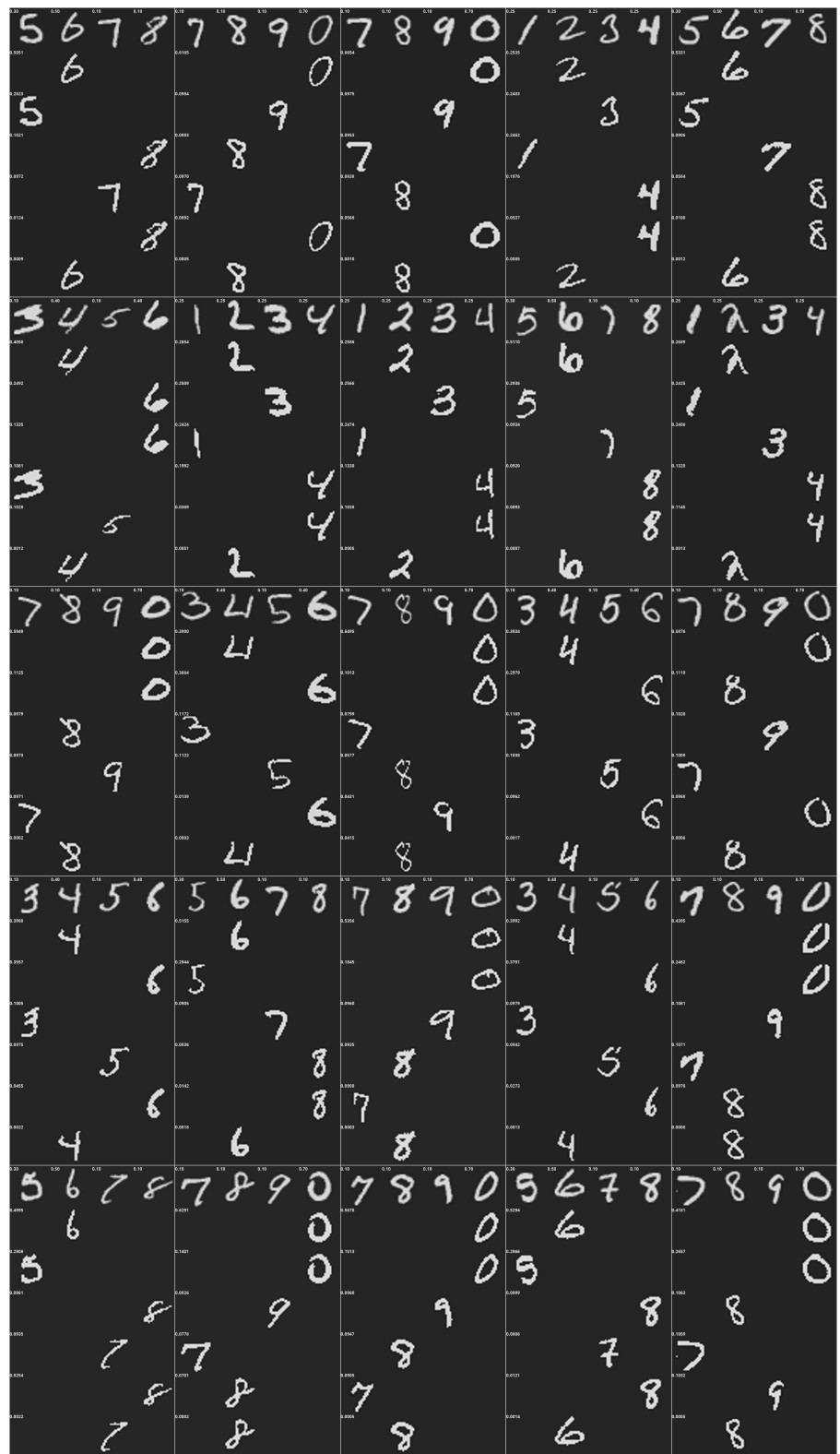

Figure 6: Results from our model on the MNIST guessing task. Top-6 results with the predicted uncertainties are shown.

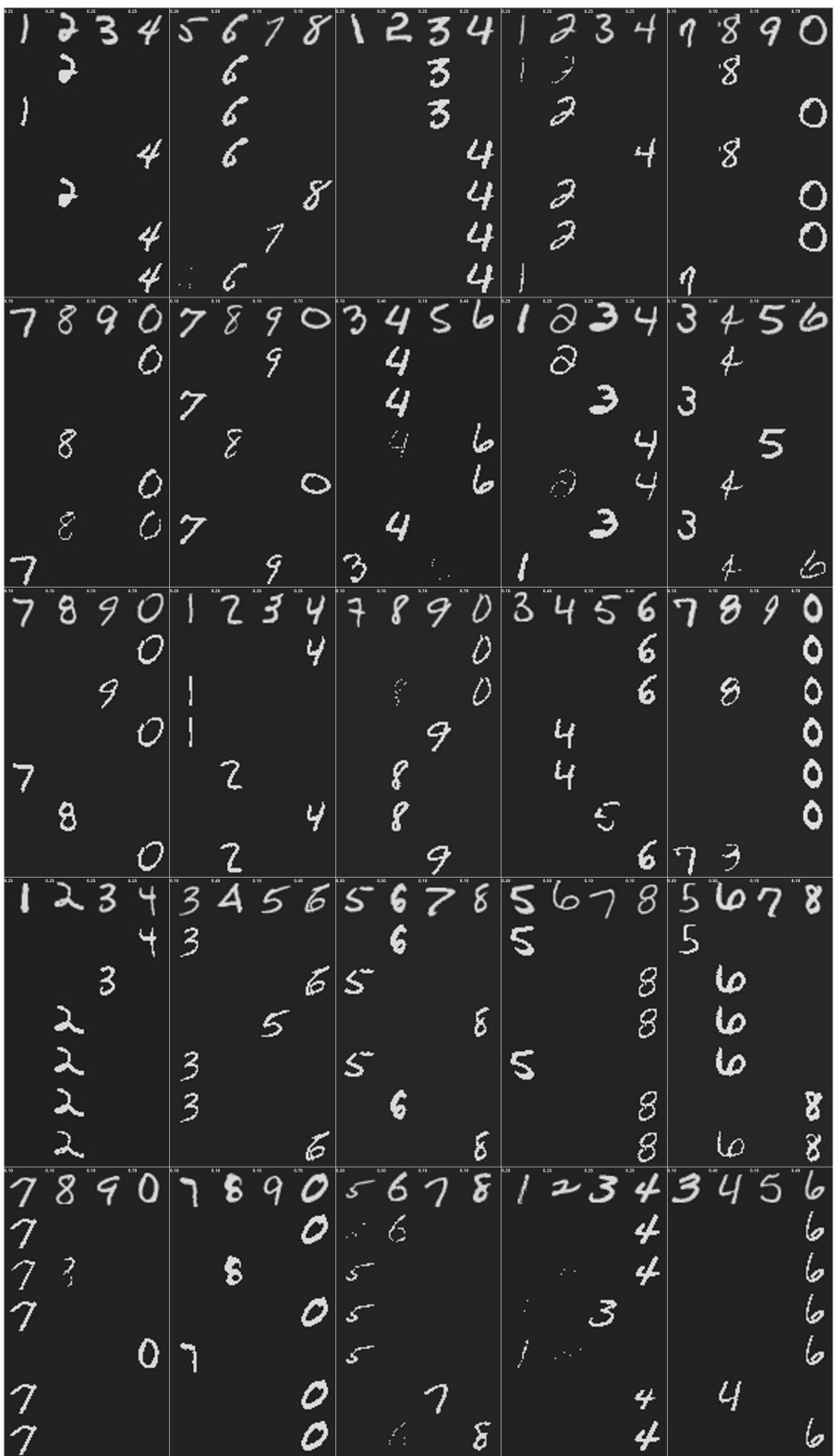

Figure 7: Results from Probabilistic U-Net on the MNIST guessing task. 6 random samples are shown.

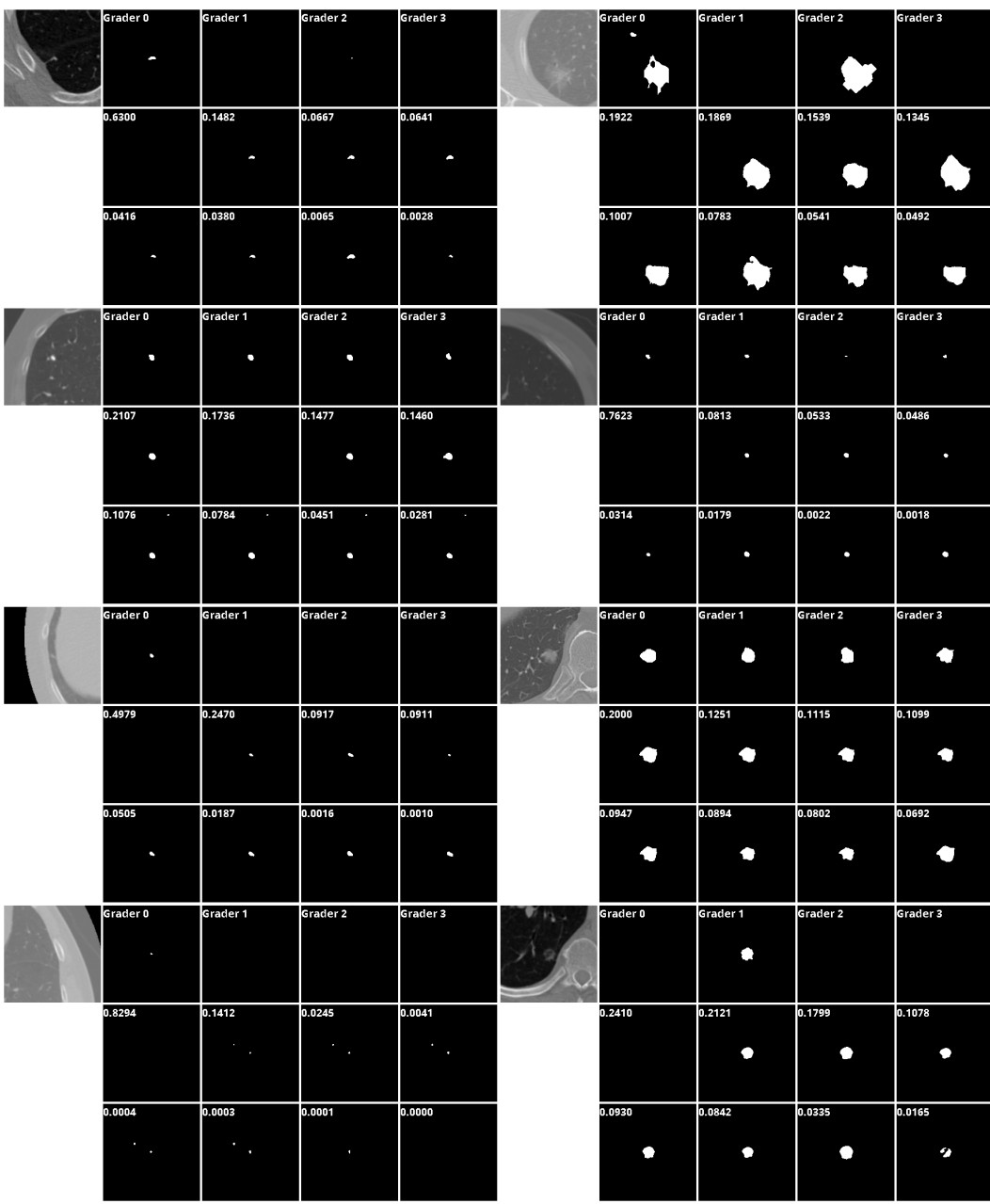

Figure 8: Results from our model on the LIDC-IDRI segmentation task. Top-8 results with the predicted uncertainties are shown.

cross entropy loss, a batch size of 256, and use the learning rate schedule $[1e^{-4}, 5e^{-5}, 1e^{-5}, 5e^{-6}]$ at $[0, 30k, 90k, 120k]$ iterations.

Some additional results from our model and Probabilistic U-Net are shown in Fig.8,9 and Fig.10, 11, respectively.

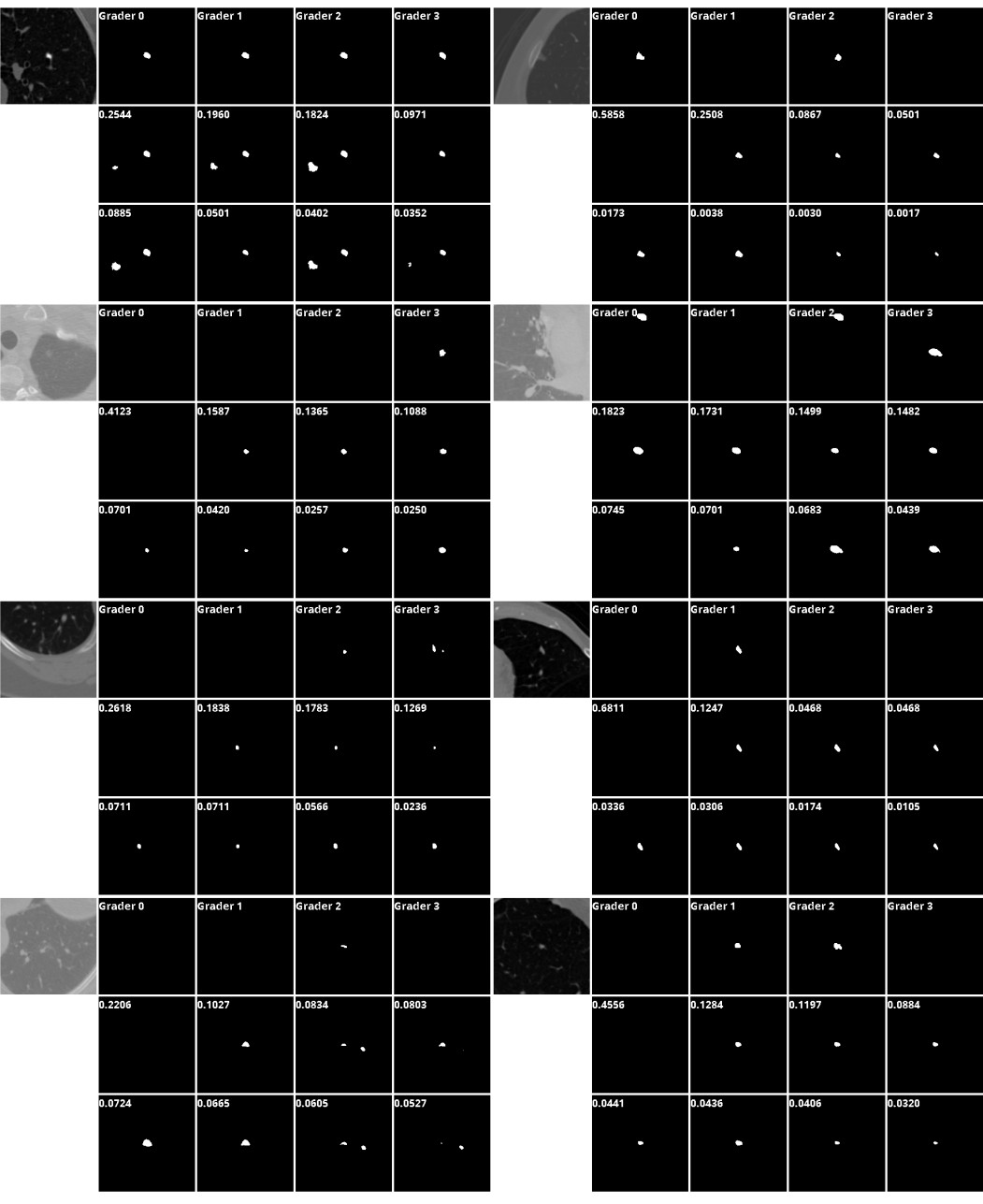

Figure 9: Results from our model on the LIDC-IDRI segmentation task. Top-8 results with the predicted uncertainties are shown.

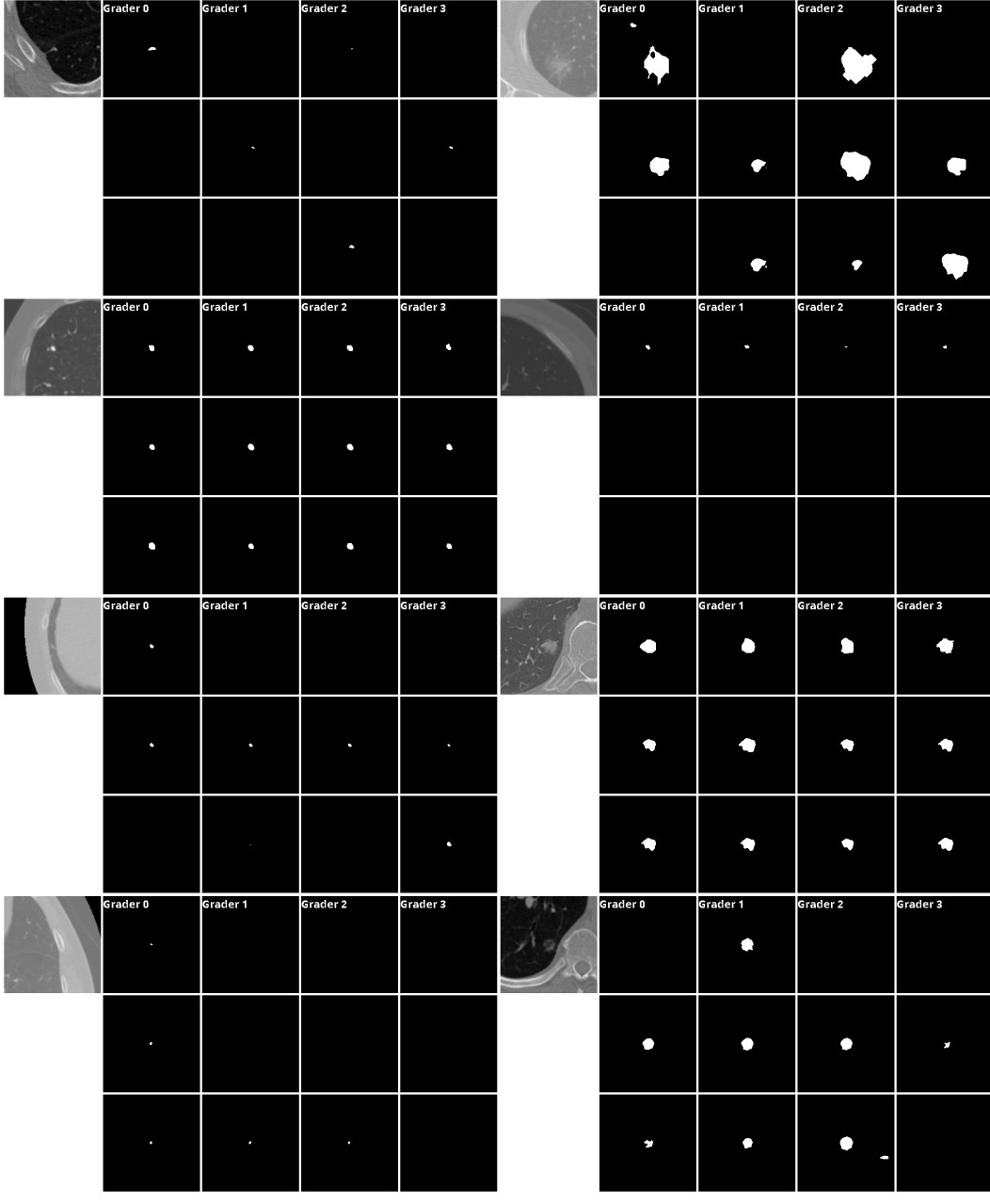

Figure 10: Results from Probabilistic U-Net on the LIDC-IDRI segmentation task. 8 random samples are shown.

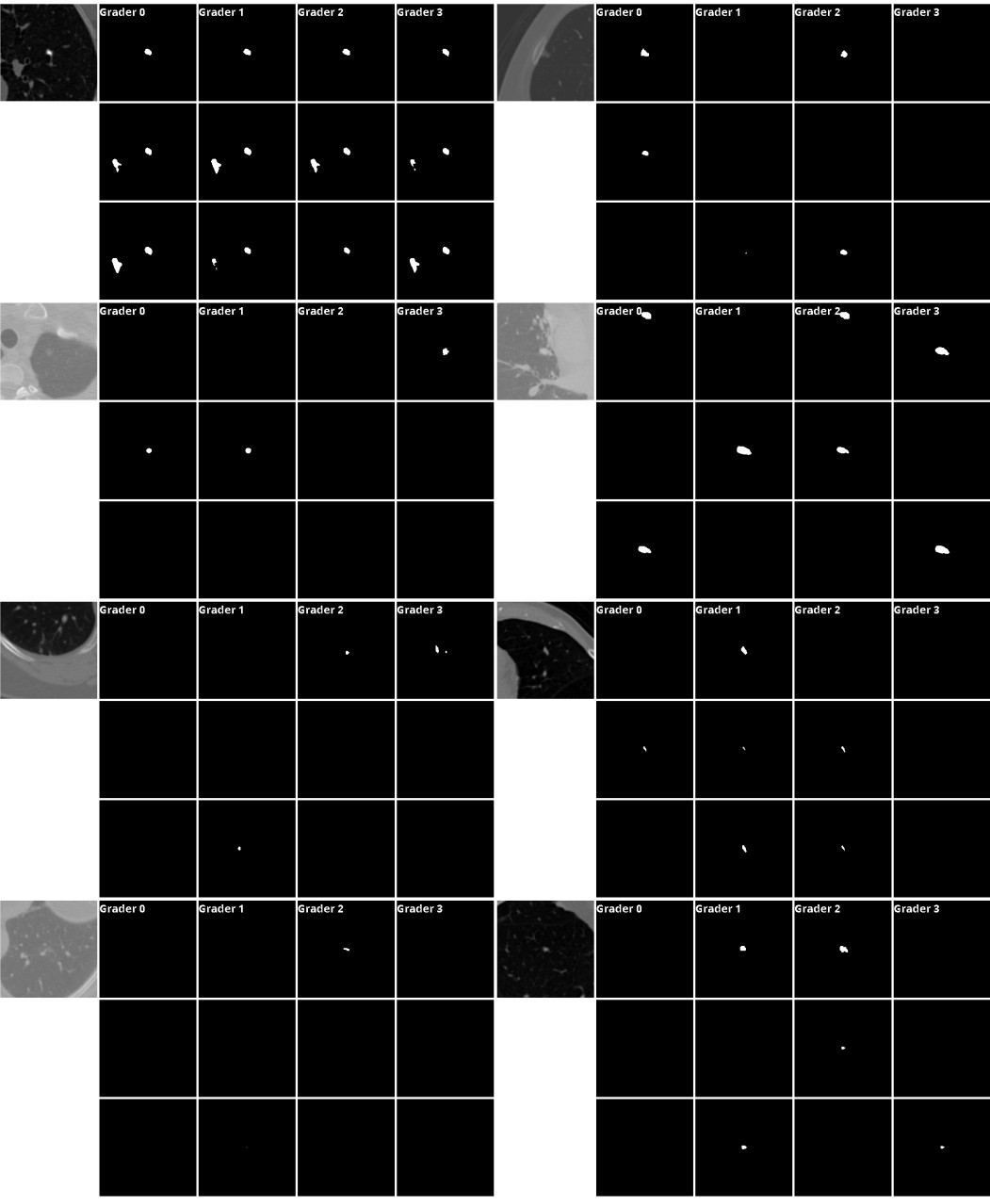

Figure 11: Results from Probabilistic U-Net on the LIDC-IDRI segmentation task. 8 random samples are shown.

