# OpenReview forum: "Modal Uncertainty Estimation via Discrete Latent Representations"
_ICLR.cc/2021/Conference — Reject_

### Official Review · AnonReviewer2 · 2020-10-25
**Extension of conditional VAE for uncertainty estimation, but need polishing**

**Rating:** 5
**Confidence:** 4

**Review:**

The paper proposes a conditional VAE like framework to learn the one-to-many mappings between input and output, leading to an application of uncertainty estimation. Technically, the novel part is to utilize a deterministic (delta) distribution for approximate posterior.

Some flaws may need future attention:

1. In the introduction paragraph staring with "Let us recall that one key ingredient of the VAE framework", the main idea is understandable: discrete latent code has definitely advantages in coping with multimodal distributions than a Gaussian distribution which is in nature single mode.  However, its explanation is confusing as following:
	In VAE, let's say we minimize KL divergence KL(P(c|x,y)|P(c|x)), where P(c|x,y) has two modes, P(c|x) has single mode. When minimization is successful, p(c|x) will spread out like figure 1(a), rather than 1(b). This is related to the difference between forward KL and reverse KL. However, it seems this paragraph suggests 1(b) as posterior collapse.

2. Using discrete latent code is not new in VAE community. There are previous works (e.g. https://arxiv.org/pdf/1804.08069.pdf) noting that naively learned discrete code c for p(y|x,c) can not be interpreted alone, but need interpreted together with input x. Such statement argues this paper's novelty and contribution.

3. Some typos and minor flaws, such as unclear references of figure rows in figure 3.

Overall, the reviewer thinks this paper need some revisions for it to be more shining.

---

> ### Author Response · Authors · 2020-11-16
> **Response**
>
> Thanks for the comments and questions! Please find below our answers.
>
> **Definition of posterior collapse**
> Thanks for the interesting question. By successful minimization, we mean that the KL divergence is small, note the situation in Fig.1(a) will have a larger KL divergence than that in Fig.1(b). Note that both prior and posterior distributions are optimized. By posterior collapse we mean negligible difference between the prior and the posterior, as is the case in Fig.1(b). The definition is consistent with the existing literature [1] cited in the paper.
>
> **Comparing with the method proposed in the neural dialogue paper**
> Thanks for bringing up the work which we were not aware of! After looking at their paper, we note there are key differences. Using our notation, we believe they would like their learned discrete code $c$ for an utterance to be "context free" at the first place in their design, meaning that the code $c$ of the response $y$ should not be influenced by the context $x$. This is in contrast to our assumption that the hypothesis $c$ of input $x$ should depend on $x$. Taking the LIDC-IDRI benchmark for an example, if we encode the hypotheses from the segmentation alone, likely there will either be two modes (benign vs malignant) or a huge number of modes if the shape of the segmentation is taken into account. Moreover, it will not contain any information about what kinds of actual biological tissue they might be, which on the other hand can be judged from the actual scan image. In our case, we have deliberately separated the recognition task learning, e.g. segmenting the image, and the hypothesis learning, so that together they can approximate the variation of $y$ given $x$. Therefore, we believe the quoted statement in their paper does not apply to our setting.
> We acknowledge that the use of discrete latent space with VAE has been explored by others, we have added in Section 2 a discussion about it. We believe that we are the first to justify the use of discrete latent space from the perspective of the multi-modal posterior collapse problem due to Gaussian parametrization of latent variables, and developed a framework to produce faithful uncertainty estimation for the one-to-many mapping problem.
>
> [1] Ali Razavi, Aaron van den Oord, Ben Poole, and Oriol Vinyals.  Preventing posterior collapse with delta-vaes. In International Conference on Learning Representations, 2018.

---

### Official Review · AnonReviewer3 · 2020-10-28
**An interesting way to apply discrete latent space VAEs, for multimodal outputs.**

**Rating:** 6
**Confidence:** 4

**Review:**

This paper introduces a novel conditional generative model for high dimensional data with multimodal output distributions. The proposed method, called modal uncertainty estimation (MUE), is a conditional VAE but with discrete latent representations. This discrete latent space allows the model to better handle multimodal outputs and provide confidence scores for the different modes predicted by the model. These capabilities are applied to the task of segmenting lesions in medical scans.


################################################

Strong points:

- The paper is clear and easy to follow. It is well-motivated and does a good job at highlighting the multi-modal posterior collapse problem.

- The model outperforms the prior state-of-the-art on both a synthetic and realistic task.


Weaknesses:

- Although the application is very different, the proposed model is very similar, albeit lighter, to a prior work (cited by the authors).

- In the shown samples, for a given input, many of the different outputs seem very similar and could be considered from the same mode. This can potentially make interpreting the probabilities more difficult than claimed in the paper, especially since the model is trained with a large number of codes. A very plausible scenario could be that one of the most likely modes is split between multiple low probability outputs and thus doesn't show up on the top ouput. Can the authors comment on this potential issue?


################################################

Score motivation:

While the method is not particularly novel, the authors apply it in a way that could be of interest to the community.
Besides, the behavior of VAEs with discrete latent space is a relevant topic that is little explored in the literature.


################################################

Other question:

I am curious about the properties of the latent space succinctly mentioned at the end of section 4. What other properties have the authors observed? Also, do similar outputs tend to be represented with codes that are close in the latent space even in the case of discrete representations?


Minor typo:

3rd paragraph of the introduction: two consecutive commas

---

> ### Author Response · Authors · 2020-11-16
> **Response**
>
> Thanks for the comments and questions! Please refer to our answers below.
>
> **On single mode splitting into multiple codes and thus affecting uncertainty estimation**
> This is a very good question!  Indeed we were concerned about the scenario where many codes correspond to the same mode, and making the uncertainty estimation less useful.  However, we found in both of our synthetic and real experiments that the extent of such a scenario is very mild. This could be explained by the fact that when it starts training, usually only one code is used, and then the number gradually increases, and then decreases. We believe there will be an interesting theory behind, in the spirit of the information bottleneck theory developed by Tishby et al. We left the theoretical aspects for the future work.  We have added corresponding discussion and references.
>
> **"I am curious about the properties of the latent space succinctly mentioned at the end of section 4. What other properties have the authors observed?"**
> We believe there are more interesting observations if we go in the reverse direction to see what kind of outputs a particular code can generate. But in order to develop a metric it requires us to have more information or professional knowledge about the scan and the patient, which unfortunately we don’t have. We believe our methodology may be of interest to people with various applications in mind.
>
> **Do similar outputs tend to be represented with codes that are close in the latent space even in the case of discrete representations?**
> We confirm that for the same image, codes that are close in the latent space produce similar outputs, since the code is constantly being updated while the results do not change significantly during the later phase of training.  We do not rule out the possibility where different codes produce similar outputs.

---

### Official Review · AnonReviewer4 · 2020-10-28
**Review 4**

**Rating:** 5
**Confidence:** 4

**Review:**

This manuscript proposes to measure the "modal uncertainty" in conditional generative models by forcing a discrete latent intermediate representation (here, C), between inputs X and outputs Y. By then manipulating the estimated categorical distribution likelihoods, an uncertainty estimate can be produced.

This is, in essence, the kludge of the VQ-VAE (van den Oord et al. 2017) categorical latent variable model into the Probabilistic U-Net (Kohl et al. 2018), which had previously used the Gaussian latent variable model. Empirically this difference is important, as the Gaussian VAE specifies a single mode latent distribution, which is incorrect in many cases.

Strong points:
*Presents a clear argument for VQ-VAE style latents vs. Gaussians (...non-unimodal uncertainty representation).
*Constructs and executes examples from a subset of previous literature (LeCun & Cortez 2010, Kohl et al. 2018) that illustrate empirical effectiveness.
Weak points:
*Limited characterization of uncertainty (i.e. uncertainty of a label image is one-of-K, or combinations thereof, and not uncertainty w.r.t. model/training/etc).
*Possibly incorrect claims (prior-free, disentanglement, Prob. U-Net ordering, VQ-VAE sampling(?)).
*Possible limited scope from the chosen definition of "uncertainty" (no model uncertainty, no parameter uncertainty).

I think this paper is marginal for this venue. It improves on the Prob. U-Net by borrowing improvements to VAE from VQ-VAE. While it does seem to better characterize label uncertainty, it is better suited to a venue where such improvements have intrinsic importance (e.g. radiology), and moreover importance that can be measured (i.e. experiments asking whether radiologists will use Non-Top-1 segmentations?). However, it appears to be an improvement on the prior art in my opinion, and I believe it would have some interest to the community as a poster.

One overall question: are we learning uncertainty of segmentations/labels, or are we simply learning the biases of the varying raters/radiologists/human segmenters, or are these two concepts indistinguishable? Supposing the number of codes equals the number of raters, why should $p(c|x)$ differ from $p(r)$? In an imbalanced rater case, why would an ordering of $p(c|x)$ not simply reflect the likelihood of one rater or another.

This seems to speak less about uncertainty in the segmentation w.r.t. the image and model parameters, and instead speak about biases in the segmentation w.r.t. the raters. While this is clearly also important, does this capture notions of uncertainty beyond the uncertainty of who is rating a certain image? Or is that only and exactly the uncertainty that the authors are attempting to capture?

Discrete Representation and Associated Loss
----

As expected, there is significant discussion and derivation devoted to the discrete representation. This essentially mirrors the short derivations in the probabilistic u-net paper, except using a discrete likelihood.

From the loss in Eq. (4) we have 1) a distribution matching term between $p_\theta(c|x)$ and $q_\phi(c|x,y)$, 2) a reconstruction term $D_\theta$ given the output of the E(x,y) encoder and $x$ (using $\arg\min || c' - E_\phi (c|x,y)||$ for $c$), and then 3) the VQ-VAE "stop gradient" term.

In the derivation slightly higher on the page, we assumed that $q_\phi(c|x,y)$ is deterministic (i.e. $q_\phi(c|x,y)$ is $1$ for one value of $c$ and otherwise $0$). While this results in a nice cross entropy term, why is it necessary? We otherwise have a still tractable difference between $q_\phi(c|x,y)$ and $p_\theta(c|x)$, summed over $c$. Is there added benefit to the forced discretization here?

Perhaps it is useful to sample $c \sim q_\phi(c|x,y)$ or use the mode $c = \arg\min || c' - E_\phi (c|x,y)||$ (this inducing an implicit gaussian likelihood structure on the embedding space), but it's not clear from just the theory that this is true.

Why is it reasonable to always use the mode reconstruction when training a loss (again, the argmax from VQ-VAE's embedding scheme), and then look at the other categories? Why should the outputs from the non-mode categories be reasonable (up to their likelihood)? (this question is in contrast to the convergence of p(c|x) to a rater distribution p(r)).

The authors claim that as in van den Oord et al 2017 there is no shrinkage to a prior over $c$. Is the regularization term $\beta || E(x,y) - sg[c]||$ not such a prior? Doesn't this enforce a structure on the co-domain of E(x,y)? Are the embedding vectors determining $c'$s learned? If not, isn't this a prior?

[minor] The authors use the terms "prior encoder" to describe $p_{\theta}(c|x)$, a learned network, and similarly "posterior encoder" to describe $q_\phi(c|x,y)$, also a learned network. Reading the probabilistic U-Net paper, it appears that the authors of that paper also use this terminology. It is a matter of viewpoint, but I personally think these terms conflict with "the usual terminology", where $p(c)$ would be "the prior", to which we shrink $p(c|x)$ to. Further, this implies a causal order to $x$ and $y$; this is the case with the empirical examples (tumor labels on CT images, etc.), but may not always be the case.

U-Net Comparison
----

The authors claim that "Probabilistic U-Net" outputs cannot be ranked; is this actually the case? It is my understanding that the difference between the proposed method and the Prob. U-Net method in architecture is the replacement of their Gaussian latent C (in their paper, Z) with a discrete categorical C. This, alongside the different loss, contains the majority of the changes.

Could the Prob. U-Net have outputs ranked by their point-likelihoods? The C output are conditionally Gaussian, so why not use the $p(c|x)$ likelihood? Understandably, generating diverse samples from this is not as simple as querying the different codes in the proposed method, but it seems incorrect to say that the samples from the Prob. U-Net are unordered.

Other Questions
----

There are several mentions of "disentanglement". In what way are these representations disentangled? They're certainly categorical by construction, but both the outputs and the actual labels are highly correlated between different categories.

At the bottom of page 3 there is a claim about sampling from a VQ-VAE being auto-regressive? Is the particular decoder architecture in VQ-VAE important w.r.t. the theory discussion directly adjacent to that statement? Surely VQ refers to the Vector-Quantization phase, which is agnostic of the decoder architecture, auto-regressive or otherwise? This seems incorrect. Similarly, there is a claim that VQ-VAE was introduced to avoid

> noise sampling, which is a different cause than ours which usually results in blurriness

Is this the cause of blurriness (not, e.g. the L2 loss in the decoder?)? And is the introduction of VQ not the same issue the authors here are attempting to address (misspecification of a single-mode latent distribution), not the additive noise sampling?

Recommendations for improvement
----
-Remove the disentanglement sentences.
-Resolve questionable claims.
-Include an experimental case where the number of raters is larger than the number of latent categories.

Edit after Rebuttal/Response Period:
----

First, an apology to the authors that a dialogue did not occur during the response period; the authors response was prompt, and my (R4) response was not, thus they were not given an opportunity to respond this response to their response to the review (...the number of recurrences may indicate why this was not possible, given limited reviewer time resources).

I think the authors misunderstand my questions about the nature of the uncertainty they're capturing:
Is there intrinsic uncertainty in the observed phenomena (e.g. medical images of tissues), are we capturing mixture proportions of deterministic states which have been mixed due to quantization, OR is the uncertainty due to the raters, i.e. found in the labels ONLY due to differences in label generation?  OR, a third case, is this moot because it does not change the outcome?

I understand that there is no explicit modelling of raters. However, my concern was that what we are capturing is intrinsically the uncertainty due to raters, even though no actual rater indicator variable was provided. This would be analogous to learning, unsupervised, the writers of the various MNISTs digits. While for MNIST this is surely difficult due to the number of writers (and their anonymity), for medical images we will likely have a limited number of raters. Having the posterior code collapse to a rater indicator appears problematic, not a desirable outcome, and likely if any one rater has correlated outputs across samples (which seems reasonable; some raters may be more or less conservative with their tissue labeling, boundaries, better/more careful at delineating curves etc). What prevents the capture of this signal, or is this the actual variation we intend to capture in the first place?

R3 further included this interesting question in their initial review:
> In the shown samples, for a given input, many of the different outputs seem very similar and could be considered from the same mode. This can potentially make interpreting the probabilities more difficult than claimed in the paper, especially since the model is trained with a large number of codes. A very plausible scenario could be that one of the most likely modes is split between multiple low probability outputs and thus doesn't show up on the top ouput [sic]. Can the authors comment on this potential issue?
If outputs are correlated i.e. overlapping in the original image domain, should the measured uncertainty be aggregated across codes?

I disagree with the characterization of the Gaussian VAE calibration in the "Ranking Probabilistic U-NET" response section. Simply because it does not accurately fit the function (one mode vs. many) does not mean we can't evaluate the learned unimodal beliefs. Yes, it is misspecified. Does this mean the rankings are meaningless? Surely the discrete model is misspecified (there are more possible masks than codes), but the ranking is claimed to be meaningful. The discrete model may have a better fit, and make the argument that it is better specified, but this doesn't mean you _can't_ evaluate the Gaussian VAE.

I stand by my initial rating and reasoning, though I note to the AC that, given space, this could make an acceptable poster. It is, in my opinion and in gross summation, an improvement on the Prob. U-Net by way of improving the Gaussian VAE sub-model of the Prob. U-Net to the VQ-VAE. This allows for sampling from a discrete set of codes which hopefully correspond with modes of the generating distribution in the data domain, instead of sampling from a parametric density, which, while continuous, has only one local maximum.

---

> ### Author Response · Authors · 2020-11-16
> **Response**
>
> Thanks for the detailed questions! Please check below for our answers.
>
> **On characterization of uncertainty**
> We have been focusing on the uncertainty from “one-to-many mapping problem”, which originates from the ambiguous label problem of the datasets, which is different in nature from model or training uncertainty. But we note that those uncertainties could be taken into consideration by using existing approaches jointly with our framework. We have updated the paper in Section 2.
>
> **On the relationship between prior distribution and posterior distribution in MUE**
> In Section 3.1, we described the fact that the *probability values* of the posterior over the discrete latent space is not pulled to the *probability values* of any prior distribution over the discrete latent space. The reason is that we maintain a delta distribution of a given input-output pair $(x,y)$ during training, which is not affected by whatever prior distribution $p(c|x)$ over $\mathcal{C}$. Note that the discrete latent space here should be considered just as a finite set of indices, without any other structure between these indices.
> As for the regularization term, it deals with how we represent the latent feature in $\mathbb{R}^n$, so that they can be utilized well by the decoder. We have made corresponding updates in Section 3.
>
> **Rater distribution vs. MUE**
> We did not explicitly model the pair $(x, y)$ to be coming from different raters. The latent code c is designed to capture the variation of the labels $y$’s conditioned on an input $x$, as provided in the dataset.  Our framework captures that overall, what probability will the scan image be compatible with $c$.
> The number of codes and the number of raters need not be the same. In the case of LIDC-IDRI benchmark, two raters may agree (the raters share the same code) or disagree (the raters hold different codes)  with each other for a certain scan image. Such ambiguous situations happen very often since no further information of the patient is provided.
>
> **On the choice of deterministic posterior**
> Because the mapping $(x, y)$ is given to the posterior encoder, there should be no “modal uncertainty” for the posterior network.  As a result, we can let the posterior encoder produce a deterministic output e in R^n for the given input-output pair (x,y). We have added more descriptions in Section 3.
>
> **How the model can produce good results, non-mode code produces possibly reasonable output**
> MUEl can be conceptually divided into two parts. The first part is an auto-encoding part: posterior encoder + decoder; the second part is a conditional generation part: prior encoder + decoder, where the prior encoder is a classification model. The prior encoder is trained by the posterior encoder. So the model can learn a good representation of the latent code c from the successfully trained auto-encoder, and at the same time faithful uncertainty estimation by the prior encoder.
> Note that although we enforced the code of mode categories to correspond to correct outputs, we didn’t enforce what output a non-mode category should give, other than they have small probabilities, because non-mode pairs don’t appear in the training. We have added more explanations in Section 3.
>
> **Ranking Probabilistic U-Net**
> In Section 1 we explained why Gaussian latent variables density value cannot be used for calibrated uncertainty estimation. We did a comparison experiment, shown in Fig.2(b), where the left axis is the point-likelihoods of the Prob.U-Net samples. We can note that the likelihood values cannot reflect the true uncertainty level and thus the order defined by these values are essentially useless.  We have added more explanations in Section 1.
>
> **On disentangled representation**
> We agree this word may lead to some confusion and we have modified them. By disentanglement we mean learning two kinds of complementary features: one that is necessary for the recognition task and one that is necessary to explain the variation of the outputs given input.
>
> **Claims regarding vq-VAE**
> The statement about sampling depends on the application of VQ-VAE. In their original paper, they considered mainly unconditional generation tasks, and the latent codes for an image in VQ-VAE are pixel-wise, hence the joint distribution of the codes cannot be obtained directly. Our sampling is easier because of the MUE setting.
> We recognize the sentence "noise sampling... blurriness" is confusing here. We would like to claim that VQ-VAE uses discrete latent variables so that it can get rid of the noise sampling, which enables the latent variable to be more effectively utilized by the decoder and produce outputs with better visual quality. While we focused on the multi-modal posterior collapse problem, which is particular to the one-to-many mapping problem.
> We have modified them accordingly.
>
> **Terminology of prior encoding networks**
> We have adopted the standard terminology, where we added citation in corresponding places.

---

### Author Response · Authors · 2020-11-16
**Overall response to the reviewers**

Thanks for the feedback from the reviewers! Overall, we have made the following improvements in the updated manuscript.
1. We have modified and highlighted our contribution in Section 1, and added in Section 2 two more paragraphs to discuss previous works using discrete latent variables in VAE, including the one noted by Review 2, and other kind of uncertainty estimation, noted by Reviewer 4, to compare ours with them and emphasize the key differences.
1. We have added more explanation about:
    (a) How the posterior distribution is not regularized by the prior distribution and the effect of the regularization term, in Section 3.
    (b) How different parts of the model work together to give good results, in Section 3.
    (c). Why scenarios where modes split into multiple codes do not become a serious problem in our approach, in Section 3.
3. We have also improved the clarity of exposition in various places, thanks to the questions and suggestions from the reviewers.

Please refer to the individual response for more details.

---

### Decision · Program_Chairs · 2021-01-07
**Final Decision**

**Decision:**

Reject

**Comment:**

This paper introduces a conditional discrete VAE for uncertainty estimation on high-dimensional data. Reviewers found the paper borderline, and two of the three reviewers stated it doesn't meet the acceptance bar due to lack of clarity in several aspects and limited technical novelty.